# Fractal Characteristics of the Spatial Texture in Traditional Miao Villages in Qiandongnan, Guizhou, China

Lei Gong 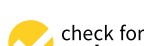, Jianzhu Yang, Chong Wu and Hui Zhou *

College of Architecture and Urban Planning, Guizhou University, Guiyang 550025, China;
gongl@gzu.edu.cn (L.G.); gs.jianzhuyang22@gzu.edu.cn (J.Y.); cwu@gzu.edu.cn (C.W.)
* Correspondence: hzhou2@gzu.edu.cn

**Abstract:** Traditional villages are the crystallization of the wisdom of human beings when living harmoniously with nature. The spatial texture of villages is complex, vague, and uncertain, making it difficult to describe with traditional Euclidean geometric spatial measurement methods. Based on the fractal theory, this study analyzes the texture of traditional Miao villages in Qiandongnan, Guizhou, China. It constructs an index system that is suitable for the fractal characteristics of traditional Miao villages. This study uses aggregation fractal dimension and capacity fractal dimension to reflect the spatial aggregation and complexity of the traditional villages, and employs the Analytic Hierarchy Process (AHP) to explore the influencing factors of spatial texture, thereby revealing the characteristics of the spatial texture of traditional villages and their relationship with the ecological environment in the process of continuous development. The research results show the following: (1) There is a significant coupling relationship among the capacity fractal dimensions of the research objects in the sampled villages, indicating that the village textures exhibit clear fractal characteristics. The villages, whether in terms of location selection or development degree, show a good fit with the surrounding environment, maintaining a relatively good original state. (2) Human factors have the greatest impact on the texture of the traditional villages, followed by natural and historical factors, economic policies, and social factors. This study provides specific development strategies for traditional Miao villages in Qiandongnan, Guizhou, offering a scientific basis for their sustainable development. It also explores a new approach for the study and conservation of the spatial morphology in traditional villages of ethnic minorities in China.

**Keywords:** Miao ethnic group; traditional villages; spatial texture; fractality; spatial morphology

## 1. Introduction

Traditional villages are living fossils that embody the adaptation of humans to the natural environment. They are rich in excellent traditional cultural heritage, such as ancient architectural techniques, ancient art, planning concepts, and the philosophy of man–land relationship [1]. Traditional villages are the crystallization of the wisdom of humans in harmonious coexistence with nature and reflect the unique cultural connotation of regional and ethnic characteristics [2]. Since the 1990s, China has undergone significant social and economic changes. Rapid industrialization and urbanization have led to the continuous flow of population, capital, and other production factors from rural areas to cities, reshaping the spatial pattern of rural areas [3]. Historic architecture in traditional Chinese villages is also facing the risk of disappearance [4]. From 2012 to 2022, the Ministry of Housing and Urban-Rural Development (MOHURD), the former Ministry of Culture (now the Ministry of Culture and Tourism, MOCT), and the Ministry of Finance (MOF) of the People's Republic of China jointly listed 8171 Chinese traditional villages in six batches for rescue and protection. These villages have received strong support in terms of funds and policies, and their living environment has been greatly improved. However, in the process of tourism development and large-scale construction, this type of cultural heritage

is facing ecological damage, homogenization of rural landscapes, and the loss of regional characteristics. This has turned the rural living environment into a "miniature bonsai" and deprived it of its profound cultural ambience [5,6]. Therefore, it is urgent to conduct in-depth and meticulous research on the spatial texture of traditional villages and their influencing factors from the perspective of overall spatial morphology.

Guizhou Province is the most densely distributed and typical area of traditional villages in China [7]. Among them, Qiandongnan of Guizhou province is the gathering place of the Miao people, and there are over 400 traditional villages. The Miao ethnic group is the oldest and largest minority group in China. According to historical records, the Miao people originated from the Yellow River region and gradually migrated to the southwest of China due to factors such as wars [7]. Therefore, they are often referred to as "mountain immigrants" and are considered one of the "most tenacious peoples", together with the Jewish people [8]. They maintain their ethnic identity through defensive landscapes and cultural heritage [9]. The uniqueness of their ethnic culture is reflected in the construction of the villages, which are vertically built based on the terrain and land capacity, exhibiting a high level of defensiveness [10]. To deal with the hilly terrain, they use the building structure to create tree-house style homes, and have come up with a versatile and diverse housing layout that fits perfectly with the mountainous landscape [11]. The spatial texture of traditional Miao villages holds special cultural significance and social functions. "Space texture" refers to the surface form composed of material space elements. From the perspective of the morphological characteristics, the space texture reflects reality by the common characteristics of groups of spatial forms. These are the dominant typing characteristics. From the perspective of element attributes, space texture is an integral part of the urban spatial form, or non-structural filled area space. Regarding the perspective of the formation mechanism, the spatial texture reflects the basic laws of the complex urban spatial structure relationship and reflects the concept and consensus that people take the initiative to create space over a certain period. From the perspective of role value, space texture as a spatial background determines the base and background color of the spatial environment and is an important source of people's perception of cities. It carries the urban historical and cultural accumulation and collective memory [12]. The continuity and inheritance of these spatial characteristics are essential prerequisites for the sustainable utilization of the cultural resources [13].

Southeast Guizhou is a typical Karst Plateau landform area, 92.5% of which is mountainous and hilly. Traffic and ecological conditions not only restrict the local economic development, but also provide a natural barrier for the preservation of historical features of villages. However, in the rapid urbanization process in China, new technologies and standards have infiltrated from cities to rural areas, damaging the original spatial form of villages [14], further leading to the convergence of village forms and landscape [15]. At the same time, rural tourism has become an important new type of development in rural China, often located in remote and conservative areas. External disturbances lead to varying degrees of erosion and damage to the ecological environment and architectural style of the village [16,17]. The sustainable development of traditional villages faces challenges, particularly in the impact on cultural landscapes.

The concept of cultural landscape was introduced by Sauer in 1925 [18]; since then, scholars have begun exploring the spatial characteristics of material cultures from the perspectives of spatial morphology and spatial occupation. As a type of material culture, the spatial morphology of traditional villages includes architectural space, street space, and overall space. Research shows that terrain, economic status, industrial status, and population density are the main factors affecting the morphology and evolution of villages [19]. The spatial morphology mainly includes natural boundaries formed by forests and water bodies, as well as artificial boundaries formed by roads, residences, and farmland. These two types of boundaries are constantly formed during development, and their interaction leads to spatial evolution [20]. In previous studies on the spatial texture of traditional villages, qualitative analysis methods were commonly adopted, and quantitative research

was relatively limited. At the macro level, shape index, boundary coefficient, aspect ratio, and other indicators are usually used to characterize the morphological characteristics of village boundaries [21,22]. The structure and differentiation characteristics of villages are reflected through Geographic Information Systems, spatial syntax, and geographic weighted regression [23–25]. At the micro level, the methods to describe the building plane combination and land use mainly include building density, dispersion coefficient, spatial gene, land use Stochastic matrix, etc. [26,27].

However, villages are influenced by a variety of natural and human factors, resulting in complex, ambiguous, and uncertain spatial boundaries and textures [28], making it difficult to describe them with traditional Euclidean geometric spatial measurement methods. Breaking through the traditional spatial measurement methods is of great significance for analyzing the morphological texture characteristics of traditional villages. Fractal geometry is a powerful tool for analyzing complex shapes in space [29,30]. When Mandelbrot analyzed the length of the British coastline, he found that the boundary length of this complex shape varied greatly with changes in the measuring scale, leading to the development of fractal theory [31]. This theory suggests that complex spatial morphologies lack scale characteristics and are difficult to characterize using traditional measures such as length, area, volume, and density [32]. Due to the scale symmetry feature of fractal objects, their complexity remains unchanged during the process of scaling up or down the measurement scale, thus represented by fractal dimensions. Currently, fractal dimensions include the Hausdorff dimension and topological dimension [33]. Subsequently, fractal theory has been widely applied in studies of urban boundaries, land use, and urban systems [34–36], but has been less explored in the research of rural spatial morphological texture [37]. This theory has unique advantages for analyzing traditional villages with obvious self-organization characteristics and complex morphology.

How can quantitative analysis of the spatial texture of traditional villages be achieved? How can researchers break through qualitative analysis and find the coupling relationship between village buildings, farmland, and ecological elements through quantitative methods? Is the corresponding relationship between the current village texture and the environment coordinated? What are the factors that affect the spatial form of traditional villages? At present, fractal morphology has formed a relatively mature theoretical system and the calculation method of fractal dimension. This study employs fractal morphology-related methods to analyze the spatial texture of traditional Miao villages in Qiandongnan, Guizhou province. This study quantitatively analyzes spatial texture through measures such as aggregation dimension and capacity dimension. It reveals the interrelationship between traditional villages and the ecological environment in the process of continuous development through quantitative data. A quantitative evaluation index system suitable for the texture of traditional Miao villages is constructed. The study employs quantitative research methods to uncover their spatial morphological characteristics and the influencing factors. Finally, specific development recommendations are proposed to provide a scientific basis for their sustainable development. Simultaneously, this study explores a new approach for the research and conservation of the spatial morphology in traditional villages of ethnic minorities in China.

## 2. Materials and Methods

### 2.1. Research Objects

The study selected Qiandongnan as the research area. Firstly, as the administrative unit with the most traditional villages in China's prefecture-level cities, Qiandongnan has a total of 415 villages listed in the Chinese Traditional Village Directory. Secondly, the ethnic basis and characteristics are obvious. Traditional villages in Qiandongnan include Miao, Dong, Han, Zhuang, Yao, Mulao, Shui, and others, with Miao villages being the most numerous and concentrated in the counties of Leishan, Congjiang, and Taijiang. Leishan serves as the center of the "Miao cultural circle". Thirdly, there is relatively well-preserved cultural heritage. Throughout its long history of development, thanks to the

natural barriers formed by its unique geographical environment, the unique folk customs, traditional craftsmanship, and regional culture of the Miao people in Qiandongnan have been well-preserved. Today, there still exist a considerable number of well-preserved Miao villages in their primitive ecological state in the Qiandongnan region of Guizhou province.

The three villages were selected from the List of Traditional Chinese Villages through the Evaluation Index System of Famous Chinese Historical and Cultural Towns and Villages (Trial) and The Evaluation Index System of Traditional Villages (Trial), and received special funds issued by China for development [38]. The selection of sample villages is shown in Figure 1. The basic information of these villages is provided in Table 1, and their spatial morphologies are described in Table 2.

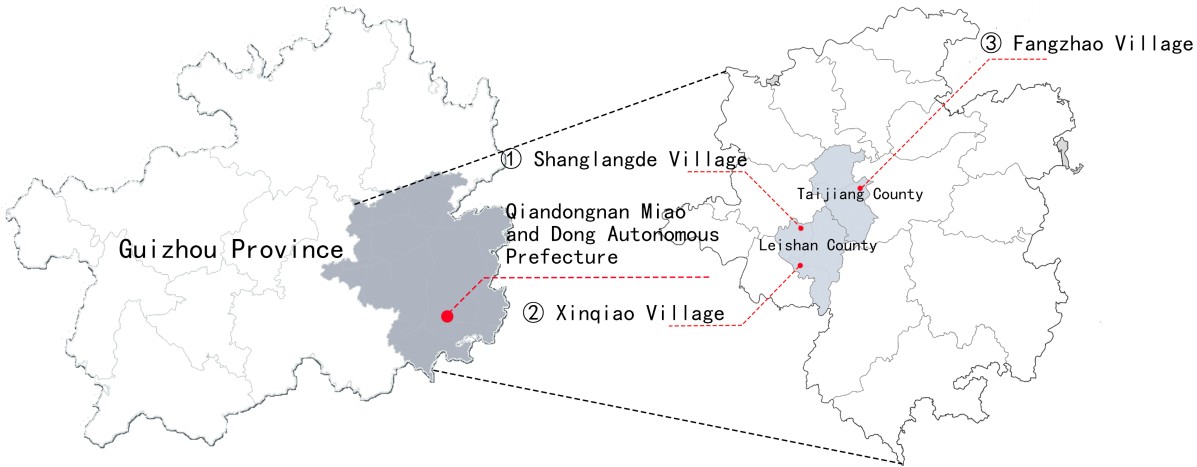

**Figure 1.** Selection of the Sample Villages. Source, author's original work.

**Table 1.** Basic Information of the Sample Villages.

| Properties | Shanglangde Village | Xinqiao Village | Fangzhao Village |
|---|---|---|---|
| County or administrative district | Langde Town, Leishan County | Datang Town, Leishan County | Fangzhao Town, Taojiang County |
| Batch | Listed in the first batch of Chinese Traditional Village Directory | Listed in the second batch of Chinese Traditional Village Directory | Listed in the fourth batch of Chinese Traditional Village Directory |
| Time of foundation | Ming Dynasty | Ming Dynasty | Ming Dynasty |
| Core Area of the village | S = 29.19 ha | S = 8.04 ha | S = 37.35 ha |
| Population | 1132 | 1020 | 2506 |
| Origin of the village | Migration | | |
| Ethnic Homogeneity | (long skirt) Miao | (short skirt) Miao | Miao |
| Representativeness | National-level cultural heritage conservation unit; national key cultural heritage conservation unit; fifth batch of China's famous historical and cultural villages. | Key tourist attraction village in Guizhou Province showcasing ethnic landscapes; core protection site for intangible cultural heritage of the Miao ethnic group in Leishan County | Distinctive and well-preserved ethnic cultural features; a typical representation of traditional villages of the region; third batch of "Villages with Distinctive Chinese Minority Characteristics" |
| Integrity | | | |
| Diversity | Located at the foot of Mount Leigong, the village is built along the mountainside, with the mountain behind and facing the water, at an altitude ranging from 735 to 1280 m. The spatial structure is characterized by clustered blocks, and the development level is high. | Spatial integrity, functional completeness, and rich environmental elements. The terrain within the village is relatively flat, with an average elevation of 910 m. The spatial structure is characterized by a strip-like pattern, and the development level is moderate. | Located at the foothills of the Mount Leigong in the Miaoling Mountain Range, the village is built along the mountainside with proximity to water. It has an average elevation of 1058 m. The spatial structure is characterized by scattered distribution, and the development level is relatively low. |

Note: Source, author's original work.

**Table 2.** Spatial Morphology of the Sample Villages.

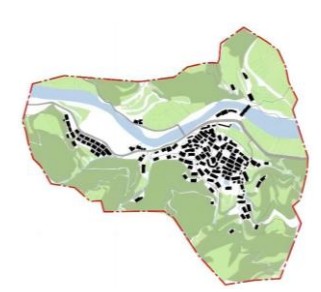 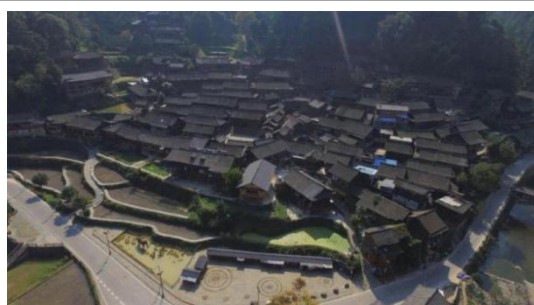

Spatial Morphology of Shanglangde Village

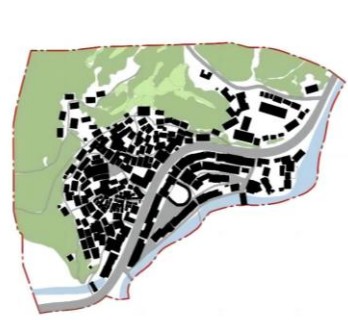 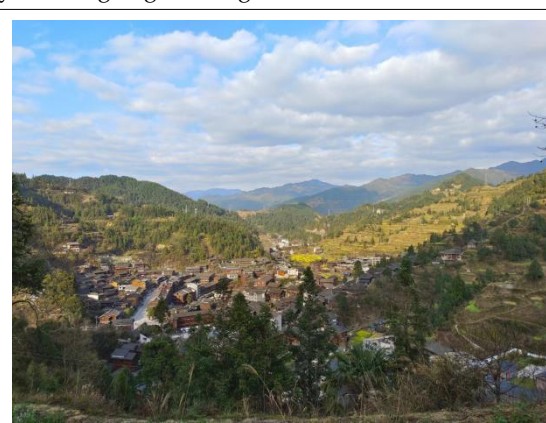

Spatial Morphology of Xinqiao Village

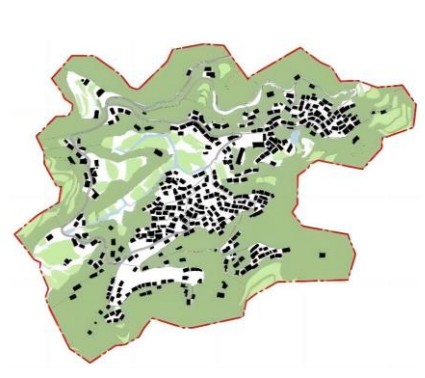 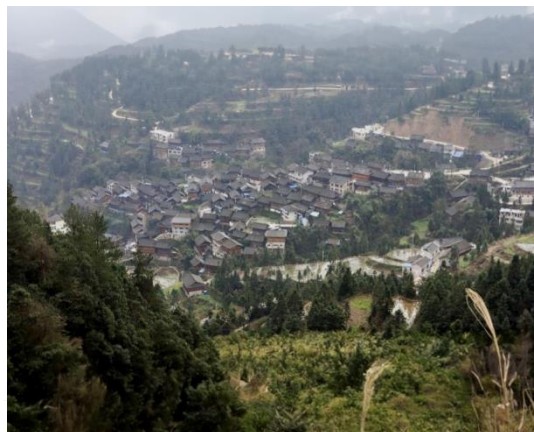

Spatial Morphology of Fangzhao Village

Note: Source, author's preparation.

### 2.2. Data Source and Collection Methods

The research team conducted field surveys at three selected case study sites from October 15 to October 30, 2020. Spatial data, including village plans and architectural layouts, were collected through drone aerial photography and on-site surveying and mapping. Questionnaire surveys and interviews with key individuals were conducted to gather relevant rural socioeconomic data. A total of 320 questionnaires were distributed, and 296 valid ones were recovered. Additionally, 16 interview transcripts were compiled as part of the research materials.

Based on the actual situation of the three sample villages, 320 questionnaires were issued using the Delphi method. We took samples according to the number of each household and rounded the last number as 5 or 0. Our interview transcripts were obtained through interviews with five to six village officials in each village. The respondents involved

are all residents of the village (including 267 ordinary villagers, 16 village officials, and 37 village teachers).

*2.3. Research Methods*

2.3.1. Fractal Dimension of Spatial Capacity

The complexity of spatial texture is represented by the capacity fractal dimension. Based on field research data and relevant surveying data, 30 different scales of grid cells with side lengths $\varepsilon$ ($\varepsilon$ = 5, 10, 20...280) were used to cover patches of various spatial elements in the traditional villages. Through geometric similarity transformations and the "covering method", the number of samples in each grid, the number of small squares with samples, and the total number of samples were statistically counted. The corresponding ln$\varepsilon$ values were calculated, and a Cartesian coordinate system was established with ln$\varepsilon$ as the $X$-axis and ln$N(\varepsilon)$ as the $Y$-axis. A double logarithmic plot of ln$\varepsilon$-ln$N(\varepsilon)$ was generated, and a straight line was fitted to the plot. The slope of the line represents the capacity fractal dimension of the patches. Finally, through pairwise comparisons of the capacity fractal dimensions of each patch in the three villages, the complexity of spatial texture in traditional Miao villages was revealed. The calculation method of the spatial capacity fractal dimension is as follows: Let F be a bounded point set on the plane, and cover F completely with a rectangle. Then, divide the rectangle into several small squares of side length $\varepsilon$ and count the number of small squares containing F, denoted as $N(\varepsilon)$. The capacity fractal dimension of F is expressed as follows:

$$D_c(F) = \lim_{n \to \infty} \frac{\ln N(\varepsilon)}{\ln N(1/\varepsilon)} \tag{1}$$

where F is not limited to a plane point set, it can be a point set on a line or a bounded point set in Rn, with the corresponding covering shape being a line segment or a cube. The following describes the method for approximately calculating the capacity fractal dimension of a point set F: Taking a plane point set F as an example, start by selecting a rectangle with a side length of a that completely covers F. Then, divide the covering rectangle into several small squares of side length $\varepsilon$ and count the number of small squares that contain F, denoted as $N(\varepsilon)$. Vary the side length of the small squares to obtain multidimensional data ($\varepsilon$, $N(\varepsilon)$). Finally, draw the scatter plot of $\ln(1/\varepsilon) - \ln N(\varepsilon)$ and fit a straight line to it, where the slope of the line is the capacity fractal dimension [39].

The analysis of the spatial occupancy and morphological diversity of traditional villages through the capacity fractal dimension can objectively reflect the spatial stability and complexity of the research object; it can be represented by the numerical value of the capacity fractal dimension (theoretical value is between 1.0 and 2.0). Mandelbrot argued that if a straight line or a plane is absolute with Euclidean dimension 1 or 2, respectively, then spatial objects such as coastlines that twist in the plane must intuitively have a fractal dimension between 1 and 2 [40–43]. When the value approaches 1.5, it indicates increasing instability in the village morphology; when the value equals 1.5, it suggests that the village patches are in a random state resembling Brownian motion, demonstrating a better fit with the surrounding environment.

2.3.2. Aggregation Fractal Dimension

The degree of spatial aggregation can be represented by the aggregation fractal dimension. A smaller aggregation fractal dimension indicates a lower level of aggregation, with spatial elements being scattered. Conversely, a higher aggregation fractal dimension indicates a higher level of aggregation and stronger integrity of spatial elements. By using the aggregation fractal dimension to describe the texture characteristics of settlements, as well as the surrounding farmland and forests of the same area, it becomes possible to accurately and comprehensively describe the sample textures based on certain scales and hierarchies. This approach reveals the texture relationship between the settlements and the

surrounding natural environment, providing quantitative data for the study of the texture characteristics of traditional Miao villages.

Based on the generated maps of settlements, surrounding farmland, and forests, the grid relationship between sample texture matrices and patches can be established using the sliding grid method, allowing for the calculation of the aggregation fractal dimension of each patch. By comparing and analyzing the aggregation fractal dimension data of each patch, the degree of spatial aggregation of the village morphology can be determined. To obtain comprehensive and objective data, this study divides the observation scale into 30 scales (ranging from 5 m to 290 m) based on the area, farmland, and architecture patch size of the sample. In the same sliding grid, if multiple patch types appear and the total area of patches exceeds two-thirds of the grid, the patch type with the largest area in the grid is selected. If a small patch exceeds half of the grid area and occupies two or more grids individually, the grid with the largest area is selected [44]. Finally, to gain a deeper understanding of the village plane morphology, correlation analysis is conducted to compare the differences between different data sets, with a significance level set at $\alpha = 0.05$. The main indicators and mathematical models for texture aggregation are as follows: in the study area, the research objects (patches) are designated as 1, and the other (matrix) is designated as background 0. The larger the gap area (area occupied by 0), the higher the aggregation fractal dimension. This analysis method is based on the grid map, and does not require the system (research object) to be stationary; it is not affected by the boundary. The specific calculation is as follows: set up a landscape map (grid map) with a side length of M, and the side length of the small grids that make up the plane grid is 1 unit, and then place a unit square in the upper left corner of the sample range (where r is an integer greater than 1). In the grids covered by the square, record the number of grids occupied by the research objects (patches) as S. Then, move the square left and right, up and down by one grid each time until it reaches the bottom right corner of the sample range. This process yields a set of different numbers of grids covered by the moving square, denoted as n(S,r). The total number of squares with side length r that can fit within the sample range is denoted as N(r). Thus, we have

$$N(r) = (M - r + 1)^2 \qquad (2)$$

In Equation (1), M is the boundary length of the landscape map. The frequency distribution of n(S,r) is converted into a probability distribution Q(S,r).

$$Q(S,r) = \frac{n(S,r)}{N(r)} \qquad (3)$$

The mean square deviation and variance of Q(S,r) are, respectively,

$$Z^{(1)} = \sum SQ(S,r) = S(r) \qquad (4)$$

$$Z^{(2)} = \sum S^2 Q(S,r) = S_s^2(r) + S^2(r) \qquad (5)$$

Mandelbrot defined gap degree as

$$\wedge(r) = \frac{Z^{(2)}}{\left(Z^{(1)}\right)^2} = \frac{S_s^2(r)}{S^2(r)} + 1 \qquad (6)$$

According to Equations (2)–(6), the gap degree is directly influenced by the size of the grid [45], the density of landscape patches, and their shapes. If the observed grid scale (with a constant value of r) remains the same, a larger value of $\wedge(r)$ indicates higher aggregation of the research objects, and vice versa. If different scales (by varying the value of r) are used to observe the same landscape, a set of landscape aggregation fractal dimension values can be obtained, allowing for the analysis of scale effects [44].

### 2.3.3. Analytic Hierarchy Process (AHP)

The AHP method is a combination of qualitative and quantitative, systematic, hierarchical analysis method. It solves a decision problem according to the total goal, where each layer target evaluation criteria are decomposed into different hierarchical structures. It then uses the solution judgment matrix feature vector to obtain each level of each element to determine a level element priority weight by using the weighted sum method and the final weight. It is an effective method to objectively describe people's subjective judgment. The AHP method is based on the original data, which can overcome the subjective evaluation of a single expert, and the obtained evaluation results can objectively reflect the weight of the index factors. The main process of the AHP method includes the following aspects: establishing a hierarchical structure model; conducting expert scoring; constructing a judgment matrix, hierarchical single ranking, and consistency tests; performing a hierarchical total ranking and consistency test; and obtaining total ranking weight.

In this study, the index system is based on the natural environment, human history, social production, and economic policy to analyze the influencing factors of spatial texture involving multiple levels. The AHP method can deal with the problem of the multi-level structure. The Analytic Hierarchy Process is employed to analyze the influencing factors of spatial texture. Finally, based on the comprehensive analysis of texture characteristics and the weights of influencing factors, a set of targeted and operable development recommendations is proposed.

Through the screening and analysis of the questionnaires, the framework of the influencing factors on the texture of traditional Miao villages is divided into three layers: the first layer is the target layer, which represents the influencing factors of the texture characteristics of traditional Miao villages; the second layer is the criterion layer, which consists of four primary influencing factors, namely natural environment factors, humanistic and historic factors, social production factors, and economic policy factors; the third layer is the indicator layer, comprising 32 secondary influencing factors, as shown in Table 3.

**Table 3.** Framework of Factors Influencing the Texture of Traditional Miao Villages.

| Target Layer | Criterion Layer | Indicator Layer | |
| --- | --- | --- | --- |
| | Natural environment(N) | Topography | N01 |
| | | Hydrogeology | N02 |
| | Humanistic and historic factors (H) | Patriarchal ethics | H01 |
| | | Site Selection Concept | H02 |
| | | Ethnic Customs | H03 |
| | | Religion | H04 |
| | | War Defense | H05 |
| Influencing factors of the mimicry characteristics of the texture of traditional Miao villages | Social production (S) | Social Organization Structure | S01 |
| | | Production and Lifestyle | S02 |
| | | Material Production and Circulation | S03 |
| | | Population Migration | S04 |
| | Economic policy (E) | Policies and Regulations | E01 |
| | | Economic System | E02 |
| | | Planning and Construction | E03 |
| | | Economic Activity | E04 |
| | | Industrial Structure | E05 |
| | | Tourism Development | E06 |

Note: Source, author's preparation.

2.3.4. Research Process

The research idea of this paper starts from the sample selection of traditional villages, to the current analysis of the spatial texture of the fractal theory, and finally uses the analytic hierarchy method to deeply analyze the genesis mechanism of the spatial texture of traditional villages. The specific process flow chart is shown in Figure 2.

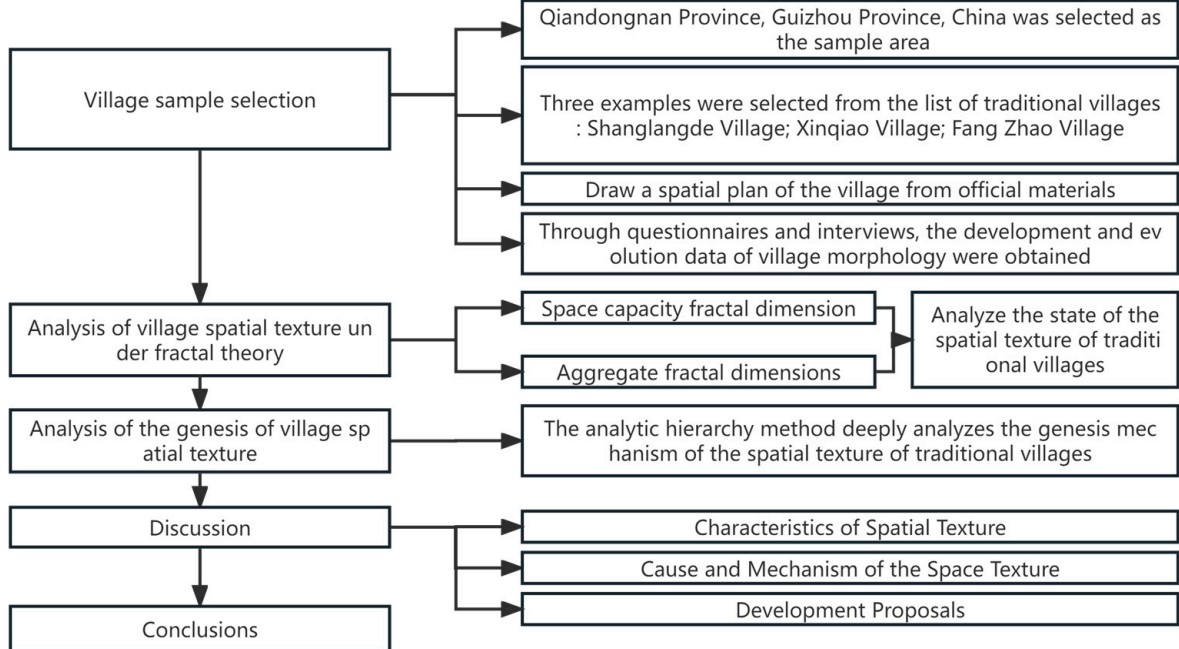

**Figure 2.** Specific process flow diagram. Source, author's original work.

**3. Results**

*3.1. Capacity Fractal Dimension*

Based on the sizes of the sample villages, 30 different grid scales were used to cover spatial elements such as settlements, farmland, and forests (ranging from 290 m, 280 m, 270 m, 260 m... down to 5 m). By counting the number of non-empty boxes for each type of element at multiple scales, the capacity fractal dimension was calculated, as well as the self-similarity among different elements. The statistics of non-empty box counts are shown in Table 4.

**Table 4.** Statistics of Non-Empty Box Counts of the Elements of the Traditional Villages.

| Side Length of the Boxes (m) | Shanglangde Village | | | Xinqiao Village | | | Fangzhao Village | | |
|---|---|---|---|---|---|---|---|---|---|
| | Non-Empty Box Counts of Settlements | Non-Empty Box Counts of Farmland | Non-Empty Box Counts of Forests | Non-Empty Box Counts of Settlements | Non-Empty Box Counts of Farmland | Non-Empty Box Counts of Forests | Non-Empty Box Counts of Settlements | Non-Empty Box Counts of Farmland | Non-Empty Box Counts of Forests |
| 290 | 8 | 9 | 10 | 4 | 2 | 3 | 9 | 10 | 10 |
| 280 | 9 | 11 | 11 | 4 | 3 | 3 | 9 | 10 | 10 |
| 270 | 10 | 11 | 11 | 4 | 4 | 3 | 10 | 10 | 11 |
| 260 | 9 | 11 | 11 | 4 | 5 | 3 | 10 | 11 | 11 |
| 250 | 11 | 11 | 11 | 5 | 6 | 4 | 10 | 11 | 11 |
| 240 | 10 | 11 | 12 | 5 | 7 | 4 | 12 | 11 | 12 |
| 230 | 12 | 12 | 12 | 5 | 8 | 4 | 14 | 12 | 14 |
| 220 | 14 | 12 | 12 | 5 | 9 | 5 | 15 | 15 | 16 |
| 210 | 12 | 13 | 12 | 7 | 10 | 6 | 14 | 16 | 17 |
| 200 | 11 | 13 | 14 | 9 | 11 | 7 | 15 | 17 | 17 |

**Table 4.** *Cont.*

| Side Length of the Boxes (m) | Shanglangde Village | | | Xinqiao Village | | | Fangzhao Village | | |
| --- | --- | --- | --- | --- | --- | --- | --- | --- | --- |
| | Non-Empty Box Counts of Settlements | Non-Empty Box Counts of Farmland | Non-Empty Box Counts of Forests | Non-Empty Box Counts of Settlements | Non-Empty Box Counts of Farmland | Non-Empty Box Counts of Forests | Non-Empty Box Counts of Settlements | Non-Empty Box Counts of Farmland | Non-Empty Box Counts of Forests |
| 190 | 14 | 14 | 14 | 7 | 12 | 8 | 15 | 18 | 19 |
| 180 | 15 | 16 | 16 | 7 | 3 | 8 | 19 | 18 | 21 |
| 170 | 16 | 17 | 17 | 7 | 3 | 8 | 18 | 19 | 22 |
| 160 | 17 | 19 | 18 | 8 | 3 | 8 | 19 | 22 | 22 |
| 150 | 21 | 23 | 22 | 7 | 3 | 9 | 23 | 23 | 27 |
| 140 | 22 | 19 | 24 | 10 | 4 | 9 | 22 | 23 | 27 |
| 130 | 23 | 27 | 26 | 12 | 4 | 9 | 30 | 23 | 32 |
| 120 | 27 | 31 | 29 | 9 | 4 | 10 | 29 | 31 | 37 |
| 110 | 33 | 36 | 36 | 10 | 4 | 11 | 34 | 33 | 43 |
| 100 | 35 | 38 | 37 | 13 | 4 | 12 | 43 | 37 | 47 |
| 90 | 45 | 46 | 44 | 15 | 4 | 13 | 49 | 37 | 57 |
| 80 | 50 | 55 | 52 | 19 | 6 | 20 | 58 | 47 | 68 |
| 70 | 64 | 64 | 61 | 24 | 6 | 21 | 67 | 55 | 84 |
| 60 | 74 | 84 | 80 | 25 | 7 | 24 | 87 | 68 | 108 |
| 50 | 101 | 108 | 107 | 35 | 7 | 32 | 123 | 84 | 149 |
| 40 | 138 | 152 | 151 | 51 | 11 | 44 | 167 | 111 | 200 |
| 30 | 212 | 233 | 237 | 78 | 16 | 70 | 264 | 162 | 332 |
| 20 | 369 | 425 | 470 | 158 | 26 | 131 | 507 | 280 | 590 |
| 10 | 1015 | 1300 | 1585 | 521 | 71 | 417 | 1507 | 723 | 1592 |
| 5 | 2780 | 4154 | 5563 | 1662 | 188 | 1384 | 4215 | 1731 | 3685 |

Note: Source, author's preparation.

As shown in Figure 3, regression analysis was performed on the logarithmic values of the sampling grid scale $\varepsilon$ and the total number of boxes $N(\varepsilon)$ for each study object. The coefficients of determination ($R^2$) for the fitted lines were all > 0.9, indicating a good linear fit and statistically significant fractal characteristics in the data. The capacity fractal dimensions of the three sample villages were all close to 1.5, indicating a random distribution and a tendency towards complexity in their overall spatial morphology. Additionally, there was a clear coupling relationship among the capacity dimensions of the studied objects in the sample villages, effectively reflecting the complexity of the texture of traditional Miao villages. The random distribution of spatial morphology in the sample villages indicates that the villages as a whole maintain a relatively pristine state and can coexist harmoniously with the surrounding natural environment.

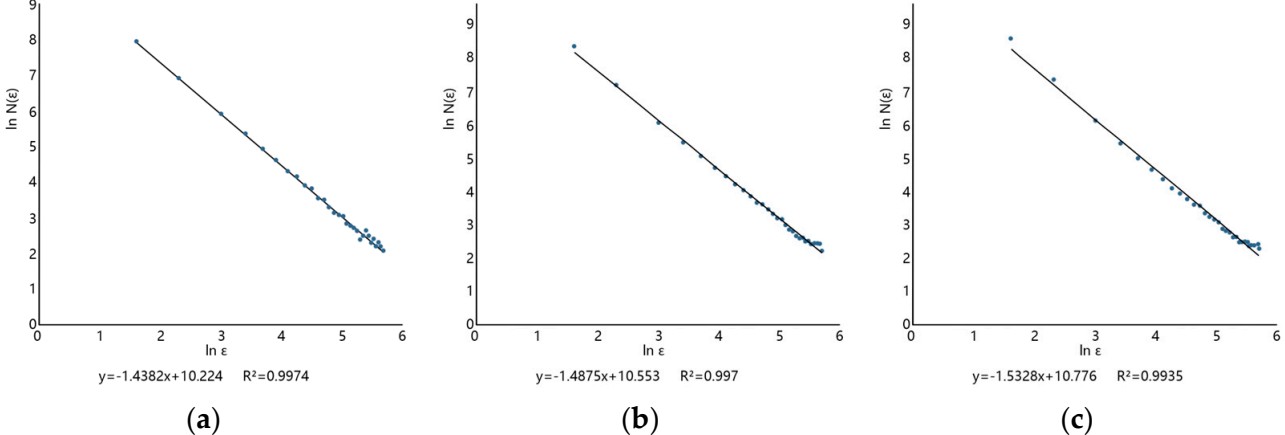

(a)   y=−1.4382x+10.224   R²=0.9974

(b)   y=−1.4875x+10.553   R²=0.997

(c)   y=−1.5328x+10.776   R²=0.9935

**Figure 3.** *Cont.*

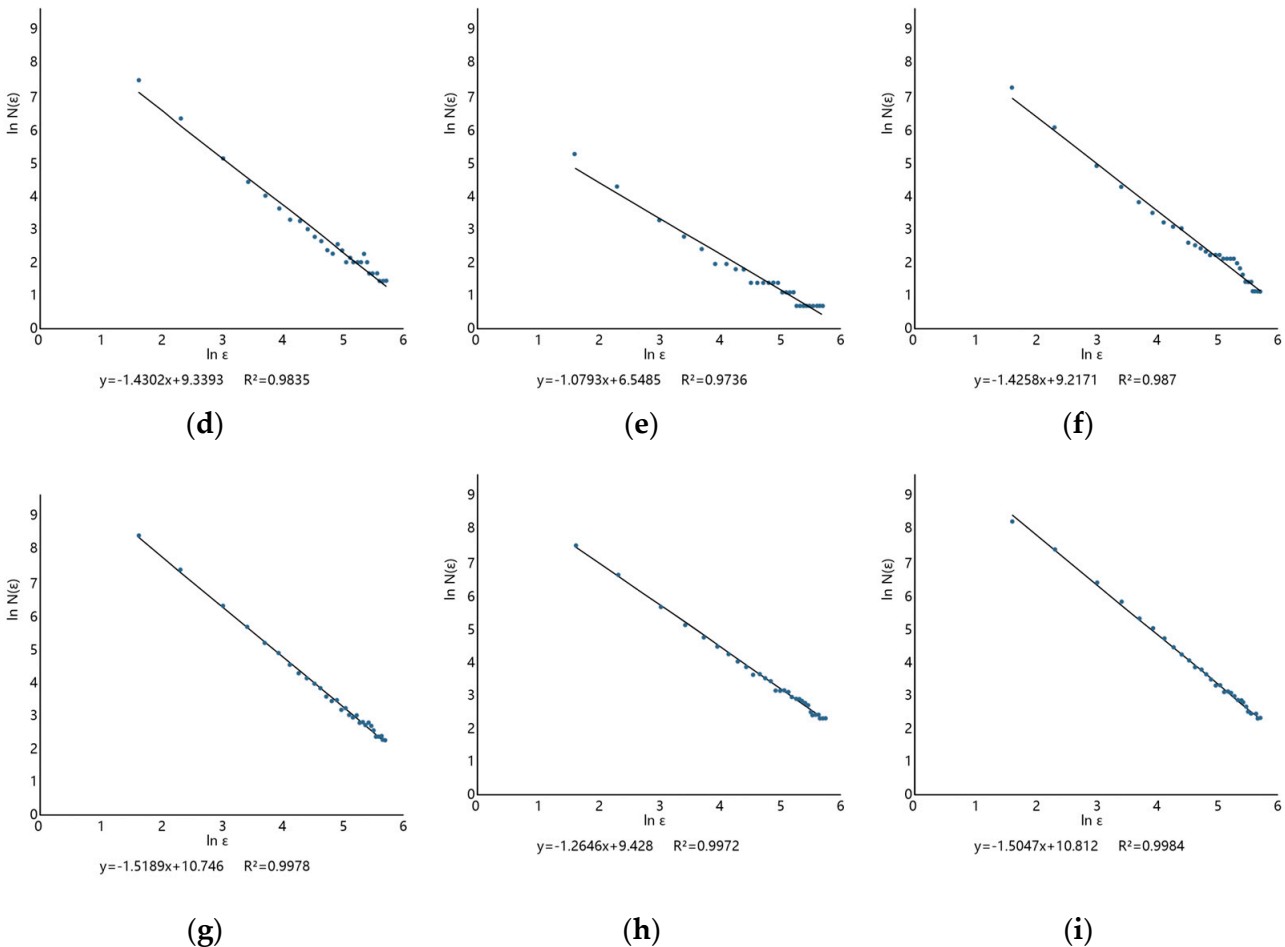

**Figure 3.** Scatter Plots of lnε-ln *N*(ε) for Sample Villages with Fitted Straight Lines. Source, author's original work. (**a**) Fitted straight line of lnε-ln *N*(ε) scatter plot for settlements in Shanglangde Village (capacity fractal dimension). (**b**) Fitted straight line of lnε-ln *N*(ε) scatter plot for farmland in Shanglangde Village (capacity fractal dimension). (**c**) Fitted straight line of lnε-ln *N*(ε) scatter plot for forests in Shanglangde Village (capacity fractal dimension). (**d**) Fitted straight line of lnε-ln *N*(ε) scatter plot for settlements in Xinqiao Village (capacity fractal dimension). (**e**) Fitted straight line of lnε-ln *N*(ε) scatter plot for farmland in Xinqiao Village (capacity fractal dimension). (**f**) Fitted straight line of lnε-ln *N*(ε) scatter plot for forests in Xinqiao Village (capacity fractal dimension). (**g**) Fitted straight line of lnε-ln *N*(ε) scatter plot for settlements in Fangzhao Village (capacity fractal dimension). (**h**) Fitted straight line of lnε-ln *N*(ε) scatter plot for farmland in Fangzhao Village (capacity fractal dimension). (**i**) Fitted straight line of lnε-ln *N*(ε) scatter plot for forests in Fangzhao Village (capacity fractal dimension).

### 3.2. Aggregation Fractal Dimension

Based on the basic data, the aggregation fractal dimensions of different patches at various observation scales ranging from 290 m to 5 m were calculated for the sample sites. The coordinates (r, ⋀) were plotted on a double logarithmic graph (as shown in Figure 4). The following conclusions can be drawn:

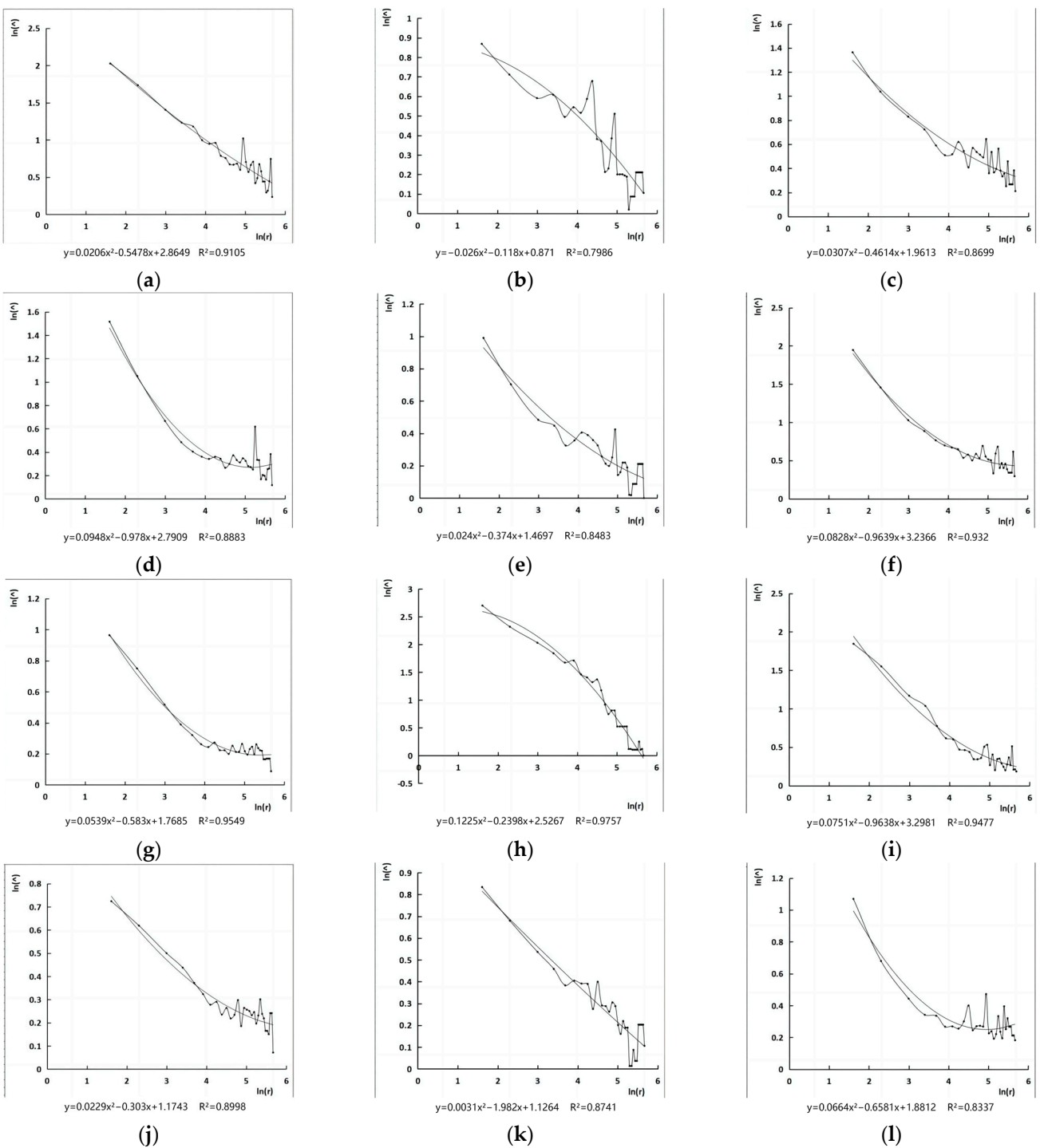

**Figure 4.** Function Graph of the Gap Degree and Sliding Grid Size of the Elements of the Sample Villages. Source, author's original work. (**a**) Aggregation fractal dimension of architecture in Shanglangde Village. (**b**) Aggregation fractal dimension of architecture in Xinqiao Village. (**c**) Aggregation fractal dimension of architecture in Fangzhao Village. (**d**) Aggregation fractal dimension of roads in Shanglangde Village. (**e**) Aggregation fractal dimension of roads in Xinqiao Village. (**f**) Aggregation fractal dimension of roads in Fangzhao Village. (**g**) Aggregation fractal dimension of farmland in Shanglangde Village. (**h**) Aggregation fractal dimension of farmland in Xinqiao Village. (**i**) Aggregation fractal dimension of farmland in Fangzhao Village. (**j**) Aggregation fractal dimension of forest in Shanglangde Village. (**k**) Aggregation fractal dimension of forest in Xinqiao Village. (**l**) Aggregation fractal dimension of forest in Fangzhao Village.

① Shanglangde Village exhibits a regular distribution pattern for its buildings, while the farmland, forests, and roads show random distributions. Furthermore, based on the calculation results at a 5 m × 5 m observation scale, the building patches in Shanglangde Village have the highest aggregation fractal dimension, followed by roads. The farmland and forests have the lowest aggregation fractal dimensions, with little difference between them. This indicates that the architectural texture of the village is the most compact, while the textures of the forests and agricultural fields demonstrate good coordination. ② Xinqiao Village shows an aggregated distribution pattern for its architectural elements and agricultural elements, a random distribution pattern for roads, and a regular distribution pattern for forests. Among these, the farmland has the highest aggregation fractal dimension, indicating a high degree of aggregation. The aggregation fractal dimensions of roads, buildings, and forests are relatively close, indicating a coordinated texture among these three elements. ③ Fangzhao Village exhibits a random distribution pattern for its buildings, roads, farmland, and forest elements. The roads have the highest aggregation fractal dimension, followed by the farmland. The aggregation fractal dimensions of buildings and forests are the smallest, with little difference between them. The data show that the road texture of the village is the most compact, while the textures of architecture and forests exhibit good coordinated aggregation.

In general, the overall analysis based on the aggregation fractal dimensions indicates that the traditional Miao villages in Qiandongnan region exhibit a random distribution pattern, reflecting a harmonious coexistence with the natural environment. The gap between the settlements and forest patches is relatively similar, suggesting a certain level of coordination between them. The aggregation fractal dimensions of architecture and roads show a positive correlation, indicating a coupling relationship between the texture aggregation of the roads and buildings in the sampled villages, with mutual influence and constraints. The texture of traditional Miao villages in Qiandongnan fits well with the surrounding environment, both in terms of the site selection and development of the villages, which is closely related to the local Miao people's ecological concept of respecting and caring for nature.

### 3.3. Analysis of Influencing Factors of the Texture Characteristics of Traditional Miao Villages

The formation, continuity, and development of any village are influenced by specific factors, including natural environment, humanistic and social factors, economic policies, and so on. The exploration of the texture of traditional Miao villages also examines the relationship between villages and their environment. It is influenced by various factors such as nature, human, society, economy, and other aspects, and is the result of the combined effect of these factors. By combining local policies, field surveys, and relevant data, the Analytic Hierarchy Process (AHP) [46] was used to analyze the weights of influencing factors and the development recommendations suitable for traditional Miao villages in Qiandongnan was proposed.

Based on the analysis results, the AHP 4.2.6 software was used to calculate the weights of the constructed judgment matrices and consistency tests were conducted. The following results were obtained, as shown in Table 5.

Based on the calculation results from Table 5, it can be determined that CR < 0.1, indicating that the judgment matrix satisfies the consistency test. From the analysis results, it can be observed that among the primary influencing factors, the most significant weight affecting the texture of traditional Miao villages is attributed to the humanistic factors, followed by natural factors, economic policies, and social factors. Among the secondary influencing factors, the top five factors are topography, site selection concepts, social organization structure, planning and construction, and production and lifestyle.

**Table 5.** Weights of Influencing Factors and Consistency Test.

| Primary Factors | Weight | Consistency Test | Secondary Factors | Weight | Consistency Test |
|---|---|---|---|---|---|
| Natural environment (N) | 0.2922 | λmax = 4.0709 CR = 0.0265 < 0.1 Test passed | Topography | 0.7500 | λmax = 2 CR = 0.0000 < 0.1 Test passed |
| | | | Hydrogeology | 0.2500 | |
| Humanistic and historic factor (H) | 0.4133 | | Patriarchal ethics | 0.2426 | λmax = 5.1825 CR = 0.0407 < 0.1 Test passed |
| | | | Site Selection Concept | 0.4678 | |
| | | | Ethnic Customs | 0.0689 | |
| | | | Religion | 0.0951 | |
| | | | War Defense | 0.1255 | |
| Social production (S) | 0.1078 | | Social Organization Structure | 0.4781 | λmax = 4.2148 CR = 0.0805 < 0.1 Test passed |
| | | | Production and Lifestyle | 0.2760 | |
| | | | Material Production and Circulation | 0.1018 | |
| | | | Population Migration | 0.1440 | |
| Economic policy (E) | 0.1867 | | Policies and Regulations | 0.1928 | λmax = 6.4262 CR = 0.0677 < 0.1 Test passed |
| | | | Economic System | 0.0992 | |
| | | | Planning and Construction | 0.3760 | |
| | | | Economic Activity | 0.0787 | |
| | | | Industrial Structure | 0.1249 | |
| | | | Tourism Development | 0.1983 | |

Note: Source, author's preparation.

This indicates that humanistic factors, particularly site selection concepts, have the greatest influence on the texture of traditional Miao villages. In Qiandongnan, with its steep mountains and complex terrain, the initial site selection plays a crucial role in determining the texture of the villages. The relationship between topography factors and site selection concepts is dialectical and indispensable in shaping the village texture. Currently, the development of traditional Miao villages is closely related to economic policies, especially in planning and construction, which impact the scale, spatial layout, village landscape, and road structure of traditional Miao villages. Additionally, it is important to note that social factors have a significant impact on the early development and construction of traditional villages, directly affecting the overall layout, spatial morphology, and the relationship with the natural ecosystem of the village.

## 4. Discussion

### 4.1. Characteristics of Spatial Texture

Because the village form and its development show unique fractal characteristics, that is, irregularities, they are scale-independent and self-similar in a certain range of scale. Therefore, it is appropriate to treat traditional villages as fractal and study their spatial forms using fractal geometry. Through the data analysis of this study, the 3-dimensional morphological and texture characteristics of traditional villages are obtained. The obtained conclusions are compared with the traditional analysis of fractal theory applied in cities.

(1)  The complexity characteristics of the texture

The three-dimensional morphological and textural characteristics of traditional villages, as well as the evolutionary process of their three-dimensional morphology and texture, can be represented through a parameterized analysis based on Euclidean geometry [47,48]. In previous studies on cities, because the urban texture is not complicated, the obtained results obtained are often related to the space itself and do not involve the spatial plane elements [49]. Compared with the study of cities, the results of the geometric fractal study of villages emphasize the complexity of texture and emphasize the fractal dimensions of different elements (forest, farmland, settlement) in villages,

so as to analyze the spatial form of villages more comprehensively. The results of this study also reveal the uncertainty and randomness in the evolution of traditional villages different from cities. Moreover, the relationship between villages and forests is closer than that of cities, and there are also rare- or even no-influence factors in cities such as farmland.

With fractal theory and research methods, the characteristics of the texture complexity in traditional Miao villages are revealed, as well as the uncertainty and randomness in the evolutionary process of these villages [50]. Overall, there is a coupling relationship among the capacity fractal dimensions of the elements in the sample villages. The forest in Shanglangde Village has the largest capacity fractal dimension, followed by farmland, and the settlement has the smallest dimension. However, the numerical difference in the capacity fractal dimensions of these three elements is relatively small, indicating a certain self-similarity among the research objects in Shanglangde Village, with a complex and evenly distributed overall spatial morphology. Due to the good preservation of the natural environment of Shanglangde village, the forest exhibits the highest capacity fractal dimension, and the most complex spatial morphology. The capacity fractal dimensions of farmland, settlements, and forests in Xinqiao Village are quite different. The capacity fractal dimensions of settlements and forests are relatively close, indicating that the two have self-similarity. Based on the field survey, this is due to the flat land within the research area being mostly used for building houses, and the originally planned farming areas have also been occupied by buildings in subsequent development, leaving only a small amount of farmland in the vicinity of the village. As a result, the farmland has the smallest capacity fractal dimension and relatively weak occupying ability in space. In Fangzhao Village, the capacity fractal dimensions of the settlement and forest are relatively close, indicating a similar morphological texture between the settlement and the forest. The capacity fractal dimension of the farmland is slightly larger compared to the settlement and forest, mainly because the village terrain is mostly rugged and steep, with farmland only distributed in a few flat areas and limited in size. Therefore, overall, the farmland has the weakest occupying ability in space and the least complex morphological texture.

Through the horizontal comparison of the three samples, it is found that the capacity fractal dimensions of the research elements in Shanglangde Village have the smallest difference, indicating a high degree of self-similarity between the settlement and the surrounding environment, resulting in the most similar morphological texture. Specifically, the ranking of the capacity fractal dimensions for settlement is highest in Fangzhao Village, followed by Xinqiao Village, and is lowest in Shanglangde Village. For farmland and forest, the capacity fractal dimension ranking is highest in Fangzhao village, followed by Shanglangde Village, and is lowest in Xinqiao Village. Based on field investigation, the main reason is that Fangzhao Village has the largest area, and its research objects exhibit relatively diverse forms, resulting in the highest capacity fractal dimension of the settlement. The higher capacity fractal dimensions for the farmland and forest in Shanglangde Village are mainly due to the better ecological environment conservation. The spatial morphology of farmland and forest in Xinqiao Village is less diverse compared to the other two villages, primarily because Xinqiao village has not preserved the natural environment as well as the other villages. Therefore, in future planning, stronger measures for ecological environment control should be implemented in Xinqiao Village (Figure 5).

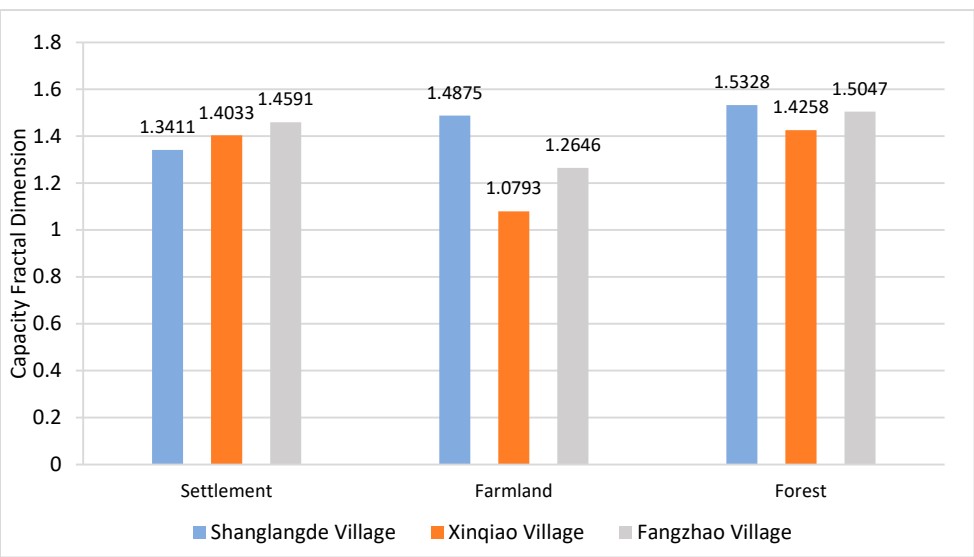

**Figure 5.** Statistics of Capacity Fractal Dimension and Scatter-point Fitted Line Coefficient of the Research Objects of Each Village. Source, author's original work.

(2) The aggregation characteristics of the texture

The correlation between the aggregation fractal dimensions of each patch in the three villages is similar, and the overall correlation is significant. This indicates that the three villages selected are generally representative, each with its own unique characteristics. The state of the natural ecological environment will affect the aggregation fractal dimensions, and there is a certain correlation between the aggregation fractal dimensions of different villages. Overall, the aggregation fractal dimensions of village buildings and village roads are relatively similar, while the gap degree of the surrounding forests is the lowest, and the farmland elements are distributed irregularly. Previous studies have shown that parametric analysis of traditional Miao villages reveals higher density in the central area and lower density in the surrounding terraces. Meanwhile, roads naturally disperse on the differential platforms [51]. This study also demonstrates similar aggregation characteristics of buildings and roads, which are adaptable to the terrain. In terms of spatial distribution, the farmland in Xinqiao village is the most densely distributed, making it the village with the tightest texture among the three. The surrounding forest of the three villages has the lowest clustering intensity, mainly because the forest surrounds the villages and is more sparsely distributed. The aggregation degree of buildings in the three villages is highest in Shanglangde Village, followed by Fangzhao Village, and lowest in Xinqiao village. In terms of the overall quality of the surrounding ecological environment, Fangzhao village ranks first, followed by Xinqiao Village, and then Shanglangde village. In general, there is a similarity in the gap degree between the settlement and forest patches. The aggregation fractal dimensions of buildings and roads show a positive correlation, indicating a coupling relationship between the roads and the texture aggregation of buildings in the sample villages, with mutual influence and mutual restriction.

The aggregation fractal dimension of the architecture patches in Shanglangde village is the highest, and is significantly different from that of the other patches. The next is the road patches, while the farmland and forest patches have the smallest aggregation fractal dimensions, with little difference between them. This indicates that the texture of the architecture is relatively tight. Based on field investigations, it is found that the village buildings in Shanglangde Village are situated according to the terrain, with a concentrated overall layout, resulting in a relatively high aggregation fractal dimension. The roads intersect between the buildings, forming a network-like distribution throughout the village and extending into the mountains outside the village, resulting in a dense spatial distribution. The village has a good ecological environment, abundant vegetation, and the

forest and farmland are nested with each other, leading to a similar aggregation fractal dimension for the forest and farmland.

The terrain of Xinqiao Village is relatively flat, with an average elevation of 910 m. The buildings are centered around water granaries and Lusheng Square, which are closely connected and centripetal, resulting in a clustered distribution of architectural elements. Although there are fewer farmland patches within the research area due to topographical reasons, they are mainly concentrated on the northeast slopes of the village, hence an aggregated distribution of farmland elements. The roads in Xinqiao village are densely woven between the buildings and fields and forests, providing convenient transportation, but without a clear pattern, resulting in a random distribution of road elements. The vegetation within the research area is nested inside and outside the village, resulting in a relatively uniform distribution pattern.

The aggregation fractal dimension of road elements in Fangzhao Village is the largest, which is significantly different from other patch types. The next is the farmland, while the aggregation fractal dimensions of buildings and forests are the smallest, with little difference between them. The data indicate that the texture morphology of its roads is relatively tight. Based on field investigations, it is found that Fangzhao village has complex terrain and lush vegetation. The buildings are scattered among the forests, forming two clusters: Dazhai and Xiaozhai. Therefore, the texture of buildings and forests are the loosest. The roads in Fangzhao Village are distributed around Dazhai and Xiaozhai in a network among settlements and their texture is relatively tight. Farmland is concentrated on flat terrain and areas near water, presenting an aggregated distribution (Figure 6).

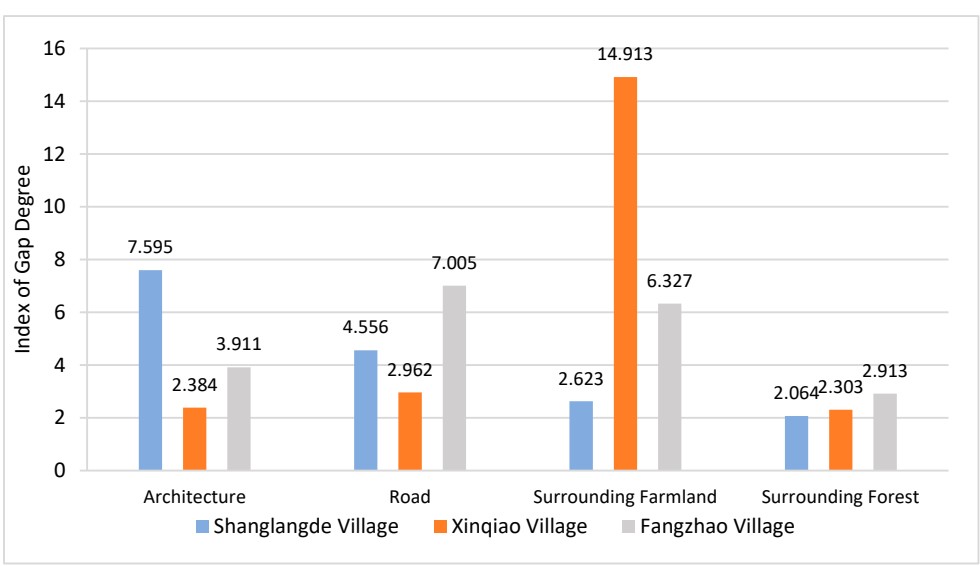

**Figure 6.** Statistics of Aggregation Fractal Dimension of Research Objects in Each Village. Source, author's original work.

*4.2. Cause and Mechanism of the Space Texture*

Overall, the morphology distribution of traditional Miao villages in Qiandongnan is in harmony with the surrounding natural environment. The selection and development of village sites align with the environment, reflecting Miao people's values of cherishing and respecting nature. The formation and evolution of the traditional villages are influenced by both natural and human factors. The value of traditional villages lies in the authenticity of architectural forms and building materials. The formation and evolution of architectural and village texture are influenced by various local natural factors such as policies, rivers, topography, and climate [50,52,53]. Additionally, the inheritance of traditional culture, beliefs, village economy, and industries also impact the spatial patterns of traditional villages [54–56].

The architectural elements of traditional Miao villages are mostly randomly distributed, which is closely related to the natural environment. Most of the villages are located on steep mountains, and buildings are randomly distributed according to the terrain along contour lines, resulting in a layout that is concentrated in most areas while dispersed in others. The fragmentation degree of land is high, and the distances between buildings are relatively short, creating a spatial morphology that is both random and ordered [48]. The road elements in traditional Miao villages exhibit a random distribution pattern. Generally, one or two main roads run through the village, while branch roads or alleys follow the terrain and are randomly distributed around the buildings and the surrounding environment. The distribution pattern of farmland elements is closely related to the village topography and the scale of farmland within the research area. Miao villages are usually built on steep mountains, leaving scarce flat areas for cultivation [57]. Farmland is often intertwined with buildings and forests, resulting in a random distribution pattern, as observed in villages including Shanglangde and Fangzhao. In smaller villages, due to topographical limitations, the available land for cultivation is limited, leading to concentrated farmland distribution and a trend of aggregation.

Humanistic and historic factors primarily influence the types of settlement, spatial structures, layout patterns, and architectural techniques, which subsequently affect the spatial structure and texture of the villages. Humanistic and historic factors do not directly and suddenly impact the spatial morphology of villages; their influence exhibits historical inheritance and time continuity. Different economic policies can affect the development scale, spatial layout, village landscape, road structure, etc., and indirectly influence the spatial pattern and texture of villages [58]. Social factors have a significant impact on the early stage of village construction, involving the fundamental layout, spatial structure, and relationship with nature, also affecting the spatial pattern and texture of villages. Natural environmental factors determine the site selection, settlement types, plane forms, and architectural materials of villages. Although natural environmental factors have the least impact on village textures among the four factors, they serve as the fundamental element for the formation and evolution of village spatial morphology, and have an influence on social, cultural, and economic factors.

The forest elements in traditional Miao villages are randomly distributed, which is closely associated with the natural attributes of vegetation. It is also a manifestation of the Miao people's respect and adaptation to nature. Over a long period of development, traditional Miao villages have lived in harmony with the ecological environment, where environmental elements are integral parts of Miao villages. In conclusion, the texture characteristics of traditional Miao villages are influenced by multiple factors, including natural environment, social factors, humanistic and historic factors, and economic policies. These factors interact and constrain each other during the formation of village textures. The factors influencing the textures of traditional Miao villages and their relationships are illustrated in Figure 7.

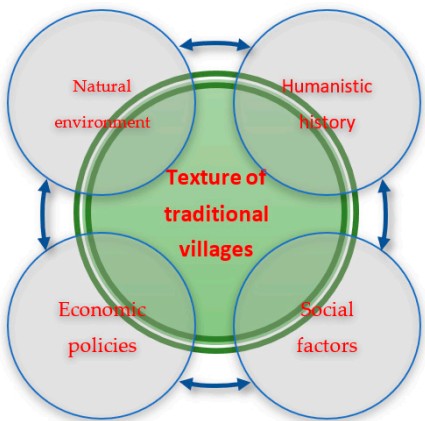

**Figure 7.** Factors Influencing the Texture of Traditional Miao Villages and Their Relationships. Source, author's original work.

*4.3. Development Proposals*

(1)  Plan for achieving harmony between humans and nature, and protect the ecological bottom line of villages

The complexity of village textures reflects the degree of coordination between villages and the natural environment. The ecological environment in Qiandongnan is generally good, with traditional villages, farmland, and forests maintaining a high level of capacity fractal dimension. However, in the construction and development of traditional villages, some villages have been relocated due to development needs. Inappropriate building volumes, village planning morphology, and ecological patterns have led to a significant decrease in their capacity fractal dimension. This not only damages the original ecological landscape but also severely impacts the continuation and protection of the rural landscape. Therefore, when traditional villages face development, it is necessary to strictly control the volume of new buildings, the morphology of village planning, and, most importantly, focus on the protection of the village's ecological environment. Only then can the harmonious coexistence between nature and the village be effectively maintained.

(2)  Respect the cultural and regional characteristics, and continue the characteristics of village textures

The research results indicate that the texture of traditional Miao villages is influenced by humanistic factors, the natural environment, and other reasons. Although they are all Miao villages, the texture and morphology of traditional Miao villages vary due to different branches, site selection, and unique topography. For example, Shanglangde Village exhibits a uniformly distributed form, Xinqiao Village is relatively flat and clustered, while Fangzhao Village has a free distribution due to its steep terrain. Therefore, in the planning process, generalization should be avoided. Instead, specific examination of their culture, history, and regional context are needed to determine the inherent logic of the settlement patterns and the natural environment, to maintain their texture characteristics so that the integrity of traditional Miao villages can be ensured.

(3)  Maintain the traditional layout of villages and rationalize the aggregation index

The research findings demonstrate that village buildings, roads, and other elements are freely and orderly distributed in accordance with contour lines and the surrounding environment. The aggregation index of village roads, buildings, farmland, and forests all have their reasonable ranges, to achieve a harmonious landscape with nature. The overall layout of traditional Miao villages in Qiandongnan exhibits a random distribution. In the process of village preservation and development, road planning should follow the topography to maintain the current level of road aggregation index.

Simultaneously, when the scale and shape of village buildings are too large, and the distribution of buildings is too concentrated, it will lead to significant changes in the architecture aggregation index. Therefore, strict control should be implemented on the distribution, quantity, and scale of the architecture, road networks, forest vegetation, farmland, and other elements so as to maintain the original pattern and continuity of the village morphology, and ensure the effective preservation of the landscape of the traditional villages.

**5. Conclusions**

The spatial texture of traditional villages has always been a focal point in academic research due to the complexity of the spatial morphology and boundaries resulting from natural environment, topography, and cultural customs. This study breaks through the previous research approach of analyzing the spatial texture of traditional villages with the Euclidean geometry method. Fractal morphology research methods are applied to analyze the spatial texture of traditional villages, using capacity fractal dimensional to analyze the complexity of traditional villages and aggregation fractal dimension to analyze the aggregation level of traditional villages. Furthermore, the Analytic Hierarchy Process

is used to analyze the influencing factors of traditional village texture. Based on the analysis results, spatial optimization proposals are put forward to provide decision-making support for the protection and sustainable development of the traditional villages. The main conclusions of the study are as follows:

(1) Regarding the complexity of the spatial texture of traditional villages in Qiandongnan, there is a significant coupling relationship between the capacity fractal dimensions of village settlements, farmland, forests, and other elements, indicating clear characteristics of complexity in the texture of traditional villages. The random distribution observed in the spatial morphology of sample villages reflects the overall preservation of a relatively pristine state of the villages and a harmonious coexistence with their surrounding natural environment.

(2) Regarding the aggregation characteristics of the texture of traditional villages in Qiandongnan, there is a significant correlation between the aggregation fractal dimensions of architecture, roads, farmland, forests, and other elements. The overall distribution of traditional Miao villages in Qiandongnan exhibits a random distribution pattern, indicating a state of harmonious integration with the natural environment. The gap degrees of the settlements and forest patches are relatively similar, suggesting a certain level of coordination between the two. The positive correlation between the aggregation fractal dimensions of buildings and roads indicates a coupling relationship between the texture aggregation of roads and buildings in sample villages, with mutual influence and restriction.

(3) Overall, the site selection and development of traditional Miao villages in Qiandongnan are in harmony with the environment, reflecting the Miao people's values of cherishing and respecting nature. The formation and evolution of traditional villages are influenced by both natural and humanistic factors.

When discussing the use of fractal theory to analyze the spatial texture of traditional villages, we must admit that this method has certain limitations in understanding the spatial morphology of villages. Fractal theory is a mathematical theory used to describe complex and irregular geometric shapes in nature. Fractal theory can be used to analyze the flat form of villages, because the spatial structure of villages often has self-similarity and hierarchy. By calculating the dimension of the village, it can reflect the spatial complexity and cultural characteristics of the village's classification. However, the fractal theory also has some limitations in analyzing the village plane form, one of which is the elevation difference aspect. Elevation level difference refers to the difference of terrain, which will affect the construction and development of villages. If only the plane form of the village is considered, and the influence of the elevation difference is ignored, then a misunderstanding of the spatial structure and cultural characteristics of the village may be caused. For example, some villages may present a lower dimension because of the steep terrain, but this does not mean that they lack cultural diversity and innovation ability. Therefore, in the subsequent analysis and research, the influence of elevation difference should be considered when discussing the cultural sustainable development of villages, so as to avoid the biased evaluation of the characteristics and value of villages. At the same time, it is also necessary to evaluate the evolutionary causes of spatial texture in traditional villages by combining the methods and indicators mentioned in this study, such as natural environment (N), cultural and historical factors (H), social production (S), and economic policy (E).

**Author Contributions:** Conceptualization, L.G., J.Y. and C.W.; methodology, L.G. and J.Y.; software, J.Y.; validation, L.G. and J.Y.; formal analysis, L.G. and J.Y.; investigation, J.Y.; resources, L.G.; data curation, L.G. and J.Y.; writing—original draft preparation, L.G. and J.Y.; writing—review and editing, L.G., C.W. and J.Y.; visualization, J.Y.; supervision, L.G.; project administration, L.G. and H.Z. All authors have read and agreed to the published version of the manuscript.

**Funding:** This research was funded by Guizhou Province Science and Technology Projects (ZK[2023]061).

**Institutional Review Board Statement:** Ethical review and approval were waived for this study, due to the following reasons: (1) the study did not involve any sensitive issues that might harm or discomfort the human participants; (2) the study anonymized or de-identified the data collected from the human participants; (3) the study complied with the ethical standards and academic norms of social science research.

**Informed Consent Statement:** All respondents signed informed consent forms.

**Data Availability Statement:** Not applicable.

**Conflicts of Interest:** The authors declare no conflict of interest.

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
