# Peer review of "Fractal Characteristics of the Spatial Texture in Traditional Miao Villages in Qiandongnan, Guizhou, China"

_sustainability, doi:10.3390/su151713218_

Round 1
Reviewer 1 Report
As the researchers of this research have reported ,the fractal theory has been widely applied in studies of urban boundaries, land use, and urban systems, but has been less explored in the research of rural spatial morphological texture. Also in previous studies on the spatial texture of traditional villages, qualitative analysis methods were commonly adopted, and quantitative research was relatively limited. For this reason, this research is very new and interesting. However, there are a few debatable points that would be useful to clarify:
1. The profile of 320 key individuals who completed the questionnaire and 16 interviewed people and how they are sampled need to be introduced. Also, including whether they live inside these villages or outside them.
2. Considering the economic and social changes that are taking place in such villages, what guarantee is there to preserve the traditional texture of these villages and in line with that, if this traditional texture is to be preserved, what economic function can be proposed for the villages.
3. Are there any laws or rules and regulations to preserve the traditional fabric of these villages in the study area?
4. In the background of the research, there have been few studies on fractal theory and spatial morphology of traditional villages, either quantitative or qualitative research, which seems to need to be strengthened.
5. Although quantitative methodology has been less used to study rural spatial morphology in the form of fractal theory, it is necessary to compare the results obtained in this research with previous studies or in other rural areas.
Reviewer 2 Report
Urban morphologists consider this topic as a fundamental area of their work. This article delves into its application through a case study of Miao Villages in Qiandongnan, Guizhou, China. There are some significant changes that need to be reviewed carefully, which include the following points:
1- The research title utilized the term "Spatial Texture," which is not commonly used in city typology. It would be more appropriate to use terms such as "urban form," "urban fabric," or "urban tissue." As the concept of "urban texture" is integral to your research, it is advisable to provide explanations of its meaning with references from established morphologists.
2 - The introduction should outline the significant challenges in the villages. This should highlight the purpose of this article. Is it to document the case or to present a framework that was ended? In both the introduction and the research design, it needed to clarify what the aim and contribution are. Is the contribution presented as factors Influencing the urban structure/urban form/ urban tissue in these villages?
3- Figure 6 outlines these factors, but it is necessary to clarify their relevance to the four key urban morphology items discussed by Conxen in the 1960s.
4- There are a few minor issues that need to be addressed. Firstly, it is important to discuss the limitations of the research in the discussion section. Additionally, it is crucial to compare the current results with those of previous studies. Finally, it is important to suggest areas for future research based on the limitations of the current study.
English is okey
Reviewer 3 Report
The paper is a great undertaking. The analyses are sound, and the discussion and conclusion are very well-written. However, the overall methodological approach of the paper is difficult to understand and could use some elaboration. Personally, I had difficulty justifying the relationship between AHP and fractal analysis.
#1: I have a question about the reference for the border of the study area for each village; were these official boundaries or drawn by the researchers? The ratio of the built area to other types seems to be influential on the outcome of the analysis, so this needs to be addressed concerning the relativity of the analysis. For example, "Xinqiao Village" is much smaller than the other two; in this case, what measures were taken to justify the comparison? (please also see comment #6). This difference is quite visible in the outcome of the study (Figure 5).
#2: The major problem with this paper is the fact that it does not establish its limitations properly. Having studied urban morphology for years, I initially assumed that the spatial characteristics of these types of rural settlements are highly influenced by topography. Your study actually shows this to be true, but the limitation is not well written.
#3: I find the item "Site Selection Concept," which, according to your analysis, is the single most influential criterion affecting the spatial characteristics of the villages, is not properly explained. What is this? How was it measured in the survey?
#4: It will be very helpful if the methodology of the paper is presented in a flowchart or diagram. It is difficult to comprehend the process through the text.
#5: The association of the capacity fractal dimension and village typologies presented in lines 178-184 requires a supportive figure.
#6: Under "3.1. Capacity Fractal Dimension," the total number of cells (boxes, occupied or not) for each "Side length of the boxes" is missing. Accordingly, it is difficult to comprehend the table because the relative proportional changes are not legible. I think it will be very beneficial if each number is supported by a ratio of the whole for that specific box size (% maybe). Please feel free to elaborate if I am mistaken.
#7: Analytic Hierarchy Process (AHP) is not explained and cited. It is not logical to employ a method without first explaining it and second, justifying why this method is suited for this analysis.
#8: The type of survey and demographic of the participants (particularly important for AHP) need to be elaborated upon.
#9: The relationship between AHP and fractal analysis needs to be further elaborated. At some points, it feels like this could become two papers.
#10: In general, the connection to the global literature is underwhelming in the paper. It is positive to cite works of scholars from the same region, but there exists a large body of international studies exploring these topics. It would increase the value of the paper to see them in the literature review or discussion.
Round 2
Reviewer 2 Report
The authors have provided a proper responses to all my concerns. Best,
Author Response
Sustainability-2501698
Fractal Characteristics of the Spatial Texture in Traditional Miao Villages in Qiandongnan, Guizhou, China
22-August-2023
Thank you for your positive and constructive comments and suggestions on our manuscript. According to the recommendation, we have made careful modification in our manuscript. The main corrections and the responds to the Reviewer’s comments are as follows:
Response to Reviewer 2 Comments
The authors have provided a proper responses to all my concerns. Best,
R: Thank you for recognizing this article. We will treat follow-up research more comprehensively and carefully.
Reviewer 3 Report
I reviewed the revised version of the manuscript and it seems that the authors have addressed all my comments.
For your future studies please do not limit your literature only to scholars of the same region and try to engage a wider range of studies. in the long run, this would only make your work stronger and better connected.
Author Response
Sustainability-2501698
Fractal Characteristics of the Spatial Texture in Traditional Miao Villages in Qiandongnan, Guizhou, China
22-August-2023
Thank you for your positive and constructive comments and suggestions on our manuscript. According to the recommendation, we have made careful modification in our manuscript. The main corrections and the responds to the Reviewer’s comments are as follows:
Response to Reviewer 3 Comments
I reviewed the revised version of the manuscript and it seems that the authors have addressed all my comments.
For your future studies please do not limit your literature only to scholars of the same region and try to engage a wider range of studies. in the long run, this would only make your work stronger and better connected.
R: Thank you very much for your positive and constructive comments on our revised manuscript. We are glad to know that you are satisfied with our revisions and that you have no further concerns or suggestions.
We appreciate your valuable advice on engaging a wider range of studies in our future research. We agree with you that this would make our work stronger and better connected. We will definitely follow your suggestion and broaden our literature review in our subsequent studies. The references we replaced this time are as follows:
24. Li T, Cao X, Qiu M, Li Y. Exploring the spatial determinants of rural poverty in the interprovincial border areas of the loess plateau in China: A village-level analysis using geographically weighted regression. ISPRS International Journal of Geo-Information. 2020, 9(6), 345.
28. Nie Z.; Li N.; Pan W.; Yang Y.; Chen W.; Hong C. Quantitative research on the form of traditional villages based on the space gene—A case study of Shibadong village in western Hunan, China. Sustainability. 2022, 14(14), 8965.
45. Liang P.; Yang X. Landscape spatial patterns in the Maowusu (Mu Us) Sandy Land, northern China and their impact factors. Catena. 2016, 145, 321-33.
Thank you again for your time and effort in reviewing our manuscript. We look forward to hearing from the editor soon.